# Agentic Discovery of Multi-Channel Bioacoustic Association Algorithms

Paramjyoti Mohapatra[1,*] Irina Tolkova[1], Akanksha Sarkar[1], Atharva Sehgal[2], Daniel P. Salisbury[1],
Andrew L. Von Duyke[3], Léa Bouffaut[1], Holger Klinck[1], Jennifer J. Sun[1]

[1]Cornell University    [2]University of Texas at Austin    [3]North Slope Borough

## Abstract

Multi-channel call association is an understudied yet essential problem in bioacoustics, serving as a critical building block for localization and density estimation. The complexity of the task stems from the fact that a robust algorithm would involve integrating spectral and temporal features across receivers. As a result, conservation researchers currently rely on time-consuming manual annotation. The challenge of searching this large, multi-constraint algorithmic space makes it an ideal testbed for agentic AI, which has recently shown a strong ability to discover novel algorithms. In this work, we investigate the effectiveness of agentic approaches for discovering generalizable algorithms for multi-channel association. We formulate multi-channel association as an agentic program synthesis problem, where candidate algorithms are discovered through iterative empirical evaluation and feedback. To address the scarcity of labeled real-world data, we rely on physics-inspired simulation and introduce a regime-scheduled evaluation and feedback strategy with warm restarts that expose candidate programs to synthetic conditions of increasing difficulty. The resulting constraint-aware algorithms generalize to unseen simulation regimes and transfer effectively to real whale call datasets, improving association accuracy.

## 1 Introduction

In a time of global biodiversity loss, marine mammals also face persistent threats – from ship strikes to climate-induced prey shifts (Avila et al., 2018; Nelms et al., 2021). Understanding the spatiotemporal distributions of threatened species is essential for effective conservation interventions (Sequeira et al., 2025). Passive acoustic monitoring has been a central methodology for studying the distribution, abundance, and behavior of marine mammals (Fleishman et al., 2023; Marques et al., 2013).

Over the past two decades, machine learning has transformed acoustic monitoring by enabling automated detection and species-level classification of animal vocalizations, substantially expanding the spatial and temporal scale of ecological analysis. However, most machine learning approaches in bioacoustics have focused on single-channel audio (Stowell, 2022). In contrast, many of the most critical conservation tasks—such as localizing and tracking individual animals—require multi-channel acoustic arrays (Rhinehart et al., 2020; Wijers et al., 2021). This introduces a significantly harder technical challenge: *multi-channel call association*. To localize a call, one must first determine which detections, recorded across a sparse array of receivers, originated from the same source. This is a complex combinatorial problem that requires jointly reasoning over learned signal similarity (e.g., spectrogram structure) and hard physical constraints such as time-difference-of-arrival (TDoA). In practice, reliable association is further complicated by noise, varying source densities, clock drift, and environmental effects.

Despite its importance, automated multi-channel association remains difficult to scale for practical use, forcing researchers to rely on time-consuming manual annotation. One class of approaches applies signal-processing heuristics tailored to specific array geometries or vocalization types, which are often brittle and difficult to generalize. Another class uses standard deep learning techniques such as metric learning, but these are typically designed for single-channel tasks and struggle to

---

*Corresponding author: pm656@cornell.edu

incorporate the hard physical constraints required for multi-channel association. Moreover, such methods depend on large-scale, manually associated datasets for each array configuration, which are largely unavailable in bioacoustics. A general and automated framework for multi-channel call association would therefore substantially streamline bioacoustic analysis.

While domain experts understand the relevant physical constraints, designing algorithms that effectively combine these constraints with spectral similarity in noisy, data-scarce settings is non-trivial. Recent work on agentic AI (Jiang et al., 2025; Toledo et al., 2025), powered by large language models (LLMs), has shown promise in addressing such challenges by iteratively searching over spaces of executable programs guided by empirical objectives and feedback. However, a key bottleneck in applying agentic program synthesis to scientific domains is evaluation: large-scale real-world datasets with ground-truth associations are not available, making it difficult to provide reliable feedback during search. Unlike prior agentic algorithm discovery work, which evaluates candidate programs under a fixed environment, we argue that in constraint-heavy scientific domains the structure of the feedback itself must evolve during search. Fixed simulators can implicitly bias discovery toward brittle heuristics that exploit artifacts of a single regime. We therefore treat simulator scheduling as a first-class design component of the program synthesis loop.

In scientific domains such as bioacoustics, scalable evaluation and feedback can be done using synthetic data generated by physics-inspired simulators. However, simulation design introduces a fundamental tension. Simulations that are too simplistic may reward brittle heuristics that exploit artifacts of synthetic data and fail to generalize, while overly complex simulations can overwhelm the search process, producing noisy or uninformative feedback.

In this work, we address this limitation by introducing a *regime-scheduled simulation orchestrator* for agentic program synthesis. Easy initial configurations help give useful feedback signals to build up a base algorithm that can be optimized further to tackle more challenging regime configurations in the later stages. The idea here is similar to that of curriculum learning in machine learning (Bengio et al., 2009).

To summarize, our contributions are as follows:

- We formulate multi-channel association as an agentic program synthesis problem, enabling automated discovery of heuristic algorithms (including adaptive preprocessing) for a core bioacoustics task without relying on large labeled real-world datasets during search.
- We introduce a regime-scheduled simulation orchestrator that evaluates candidate programs under a sequence of synthetic conditions of increasing difficulty, addressing key limitations of fixed simulator configurations in agentic program synthesis.
- Through controlled ablations, we analyze how warm starts and regime scheduling affect the diversity and stability of discovered algorithms.
- We show that algorithms discovered using regime-scheduled simulation generalize to unseen synthetic conditions and transfer meaningfully to held-out synthetic datasets and real-world bioacoustic datasets.

## 2 RELATED WORK

**Computational Bioacoustics & Source Association**  Computational bioacoustics has traditionally relied on classical signal processing and array analysis methods derived from naval sonar research (Au & Hastings, 2008). Over the past decade, deep learning has become dominant for detection and species-level classification of animal vocalizations, demonstrating strong performance across both low-frequency baleen whales and high-frequency toothed whales (Shiu et al., 2020; Kirsebom et al., 2020; Schall et al., 2024; Frasier, 2021).

Despite these advances, most methods operate on single-channel audio, while multi-channel association remains comparatively underexplored. Acoustic arrays are widely used for localization and density estimation (Sousa-Lima et al., 2013; Fleishman et al., 2023), where accurate cross-receiver call association is a prerequisite. Prior approaches typically enforce physical feasibility through TDoA rejection or combine temporal constraints with cross-correlation-based similarity measures in rule-based or metric-driven pipelines (Helble et al., 2015; Nosal, 2013; Lapp et al., 2023; Hendricks et al., 2019; Baggenstoss, 2011). However, these techniques often degrade under high call density or

require extensive manual tuning, and manual association remains common in practice (Tenorio-Hallé et al., 2022). Once association is resolved, localization is typically performed via multilateration using least-squares, Bayesian, or likelihood-based formulations (Urazghildiiev & Clark, 2013; Rideout et al., 2013).

**AI Agents for Science and Algorithm Discovery**   Recent progress in large language models has enabled agentic frameworks that iteratively generate, execute, and refine code, reducing the manual effort required to build scientific software. Such systems have demonstrated utility across domains including biology, mathematics, and materials science (Huang et al., 2025; Novikov et al., 2025; Zhang et al., 2024), forming part of a broader movement in AI for Science (Wang et al., 2023). Unlike standard predictive tasks, multi-channel bioacoustic association is a constrained symbolic problem that requires balancing learned similarity measures with hard physical constraints, making it well-suited to search-based agentic approaches.

Parallel advances in LLM code generation increasingly treat program synthesis as an iterative decision process guided by execution feedback and structured search rather than single-pass decoding (Chen, 2021; Roziere et al., 2023; Snell et al., 2024). Tree and population-based strategies—including beam search, Monte-Carlo tree search, and evolutionary refinement—have shown consistent gains on coding and reasoning benchmarks by maintaining and improving multiple candidate programs (Yao et al., 2023; Wang et al., 2022; Hazra et al., 2024).

## 3   METHOD

**Problem Setup**   We assume detections are given as input and focus exclusively on the association problem, excluding upstream detection or segmentation. Each detection includes spectrogram data together with metadata such as time of arrival, receiver location, and receiver identity. The task is to partition detections into groups corresponding to individual sources.

Effective association depends not only on grouping logic but also on adaptive preprocessing and representation of raw observations, particularly under varying noise conditions. We therefore treat preprocessing, feature extraction, and constrained grouping as a single algorithmic pipeline to be discovered. Given a candidate algorithm, predicted groupings are evaluated against ground truth using task-specific objective Hungarian-matched precision, recall, and F1 scores, while enforcing domain constraints such as physical feasibility or channel uniqueness. The exact problem description provided to the agent can be found in C.

### 3.1   AGENTIC ALGORITHM GENERATION FRAMEWORK

**Iterative Synthesis Loop**   Algorithm generation proceeds as an iterative propose–evaluate–refine process. At each step, the agent proposes a candidate `associate_calls` program conditioned on the task description and previously evaluated candidates. The candidate is executed on simulated instances and assigned a score which is then used to guide subsequent edits.

**Search Policy**   For the base search policy, we adopt a UCT-based tree search over candidate programs, building on Jiang et al. (2025). Nodes correspond to candidate programs and edges correspond to edit operations. Selection balances exploitation of high-scoring candidates with exploration of under-visited branches via a UCT criterion. More details about the policy can be found in B in the appendix.

**Evaluation Objective**   Each candidate program implements `associate_calls(detections)` and outputs a grouping for all detections in an instance. We compare predicted groupings to ground truth using Hungarian-matched, weighted F1. Alongside F1, we compute structural diagnostics (e.g., singleton fraction, duplicate-channel rate, mean group size, precision, recall, and soft TDoA violation frequency) to characterize physical plausibility and failure modes. These diagnostics are provided as textual feedback during the synthesis loop.

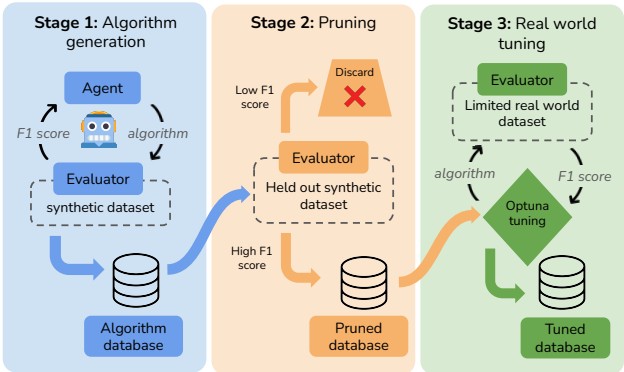

Figure 1: Three-stage agentic synthesis pipeline consisting of algorithm generation on synthetic data, pruning using evaluation on held-out synthetic data, and light parameter tuning using Optuna on target data. The first stage generates a large corpus of programs for our task, the second stage prunes out programs that fail to generalize to unseen synthetic setting, while the third stage tunes the parameters on a small subset of our target set.

## 3.2 SIMULATION-BASED EVALUATION

**Simulator** We use a physics-based simulator to generate synthetic multi-channel association problems with known ground-truth groupings. Each simulated instance consists of a set of detections with associated metadata (receiver identity, time of arrival, and spectrograms), generated by sampling latent source locations and simulating their propagation across an acoustic array.

The simulator exposes several controllable parameters that affect task difficulty while preserving the underlying physical structure of the problem. These include: (i) the number of calls (source density), (ii) the number of receivers, (iii) Odds of a call being picked up by a receiver (Det. Noise where larger values corresponds to wider detection radius), and (iv) spectrogram corruption via additive noise and masking. By varying these parameters, the simulator produces association problems with differing levels of ambiguity and noise.

**Scheduled Simulation Regimes** Rather than evaluating candidate programs under a single fixed simulator configuration, we define a sequence of pre-specified simulation regimes with increasing difficulty. Each regime corresponds to a distinct setting of simulator parameters, summarized in Table 9 in the Appendix. Early regimes emphasize simpler scenarios with lower density and noise, while later regimes progressively increase ambiguity through higher call density, varying receiver geometries, and stronger spectrogram corruption.

For each regime, we ensure that the algorithm is allowed to evolve for at least 20 and at most 60 iterations. Between 20 and 60 iterations, an algorithm is allowed to progress to next regime with if it has achieved a specified threshold objective score – this ensures that we don't waste compute to over optimize on regimes where the algorithm already has good performance.

## 3.3 WARM STARTS

When transitioning between successive simulation regimes, we employ a warm-start strategy. Rather than restarting synthesis from scratch, each new regime is initialized using the best-performing algorithm discovered in the previous regime and the problem description is modified to include a description of the best performing algorithm from all previous stages. This allows the search process to incrementally refine existing algorithmic structure as task difficulty increases. Based on previous works, we hypothesize that a combination of warm starts and schedule regimes can facilitate stable convergence and help converge to better solutions (Bengio et al., 2009).

## 4 EXPERIMENTS

Our experiments are designed to study i) how well do algorithms generated using simulated data generalize to unseen data; ii) what level of fine tuning might be needed post algorithm discovery

| E1 | Single regime / no warm start / F1-only | E2 | Single regime / warm start / F1-only |
|----|------------------------------------------|----|--------------------------------------|
| E3 | Scheduled regimes / warm start / F1-only | E4 | Scheduled regimes / warm start / composite |
| E5 | Combined regime / no warm start / F1-only | | |

Table 1: Glossary of experimental configurations. Refer§4.1 for further discussions.

for effective sim to real transfer; iii) how scheduled simulation regimes and warm-start strategies influence the quality, robustness, and diversity of algorithms discovered during agentic program synthesis. In particular, we want to analyze how algorithms generated transfer to i) Synthetic held out sets; ii) Real World Datasets. We focus first on controlled synthetic evaluation, and subsequently assess real-world transfer.

**Baselines** We compare the accuracy (weighted F1) of the generated algorithms against the following:

**B1** (DL Embedding + Clustering); **B2** (Greedy TDoA based clustering); **B3** (GPT-5.2, 1-shot algorithm extraction followed by Optuna finetuning).

## 4.1 EXPERIMENTAL CONFIGURATIONS

We consider five synthesis configurations to evaluate the impact of the following factors: (i) single regime versus scheduled simulation regimes, (ii) warm-started versus no warm-started synthesis, (iii) objective function design for algorithm ranking (F1 vs Composite score consisting of F1, precision, and singleton fraction). Namely, configurations E1-E5 helps us study those. All configurations use the same base search policy (3.1) and identical LLMs for code generation, feedback, and summarization (Gemini 2.5 Pro).

The configurations are presented in Table 1. Each configuration is run for three independent synthesis runs with different random seeds to capture the stochasticity of the search process.

## 4.2 SYNTHETIC EVOLUTION AND HELD-OUT EVALUATION

During synthesis, candidate algorithms are evaluated exclusively on synthetic problem instances from the current simulation regime (analogous to a training set). To avoid selection bias and overfitting to the training simulator distribution, algorithm selection is performed using a separate held-out synthetic dataset (analogous to a test set) that is not used during synthesis.

The held-out synthetic set is generated by sampling new random seeds and simulator realizations, with parameter values disjoint from those seen during synthesis.

Each synthesis run produces a collection of candidate algorithms evaluated over time. In particular, for the experiments E2, E3, and E4 with warm starts, we sample the best algorithm from the specified regimes (totaling 11 for these experiments). For E1, E5, and E6 with no levels or warm starts, we sample the top algorithm from 11 uniform bins from the entire run. We report the average F1 score of these top-5 algorithms for each run.

This protocol allows us to compare not only peak performance but also the consistency with which strong algorithms are discovered under different evaluation strategies while also demonstrating trends in performance over iterations. The results for this dataset can be found in 2.

## 4.3 REAL-WORLD EVALUATION

Following synthetic evaluation and selection, we evaluate the selected algorithms on real-world multi-channel whale call dataset with manually annotated associations collected using the pipeline described in (Garcia et al., 2025). The dataset consists of 405 source whale calls of which 33 are used for Optuna tuning (Akiba et al., 2019) and 372 for validating algorithm performance. Optuna tuning is done for 50 iterations for each algorithm. Furthermore, since the original dataset's source calls were fairly sparse over time, we also use a variant of this dataset by artificially compressing the times to a duration of 400s (60 calls / minute) for validation. The results on this dataset can be found in tables 4, 5, 6, and 7. A more detailed description of this dataset can be found in D in the Appendix.

Table 2: F1 Scores using baselines and No Synthetic Data Configuration (E0)

| Dataset | B1 | B2 | B3 |
|---|---|---|---|
| Sparse Real World Dataset | 0.34 | 0.67 | $0.72 \pm 0.2$ |
| Dense Real World Dataset | 0.34 | 0.45 | $0.47 \pm 0.17$ |

Table 3: Held-out synthetic F1 scores (Top-5) and iteration counts for each experiment (E1–E5), reported across three independent runs.

| | Run 1 | | Run 2 | | Run 3 | |
|---|---|---|---|---|---|---|
| Experiment | F1 | # Iterations | F1 | # Iterations | F1 | # Iterations |
| Config. E1 | $0.904 \pm 0.005$ | 350 | $0.526 \pm 0.192$ | 350 | $0.442 \pm 0.004$ | 350 |
| Config. E2 | $0.758 \pm 0.080$ | 660 | $0.940 \pm 0.014$ | 226 | $0.410 \pm 0.020$ | 660 |
| Config. E3 | $0.940 \pm 0.013$ | 380 | $0.744 \pm 0.017$ | 620 | $0.930 \pm 0.019$ | 380 |
| Config. E4 | $0.950 \pm 0.006$ | 225 | $0.930 \pm 0.013$ | 398 | $0.932 \pm 0.011$ | 348 |
| Config. E5 | $0.396 \pm 0.005$ | 350 | $0.490 \pm 0.190$ | 350 | $0.380 \pm 0$ | 350 |

## 5 DISCUSSION

**Do the generated algorithms generalize to real world settings?**   Our results indicate that algorithms discovered via agentic program synthesis and selection in simulated enviroments exhibit partial zero-shot transfer to real-world datasets, but consistently benefit from lightweight parameter adaptation on target sets. Applying Optuna fine-tuning yields an average improvement of approximately +0.10 F1 on the sparse real-world dataset and +0.04 F1 on the dense real-world dataset across 9 runs.

Notably, certain configurations show substantially larger gains. For instance, E3 Run 1 improves by +0.29 F1 on the sparse dataset and +0.12 F1 on the dense dataset suggesting that while the structural heuristics discovered by the agent are often sound, their performance is sensitive to parameter calibration under domain shift.

These findings suggest that agentic program synthesis appears to be effective at identifying robust algorithmic structures when exposed to a varied challenging regimes while small amounts of target-domain tuning significantly enhances practical performance.

**Are synthetic held out sets accurate indicators of performance on real world sets?**   We observe a strong positive correlation between held-out synthetic F1 and average post-tuning real-world F1 (averaged on sparse and dense real world sets) as seen in 2. This indicates that synthetic validation performance serves as a reliable ranking signal for selecting candidate algorithms prior to real-world tuning.

**Do warm starts and the scheduled regime help?**   While the experiment with warm starts and schedule regimes (E3) achieved the best F1 on average, such performance metrics given a small number of runs are highly dependent on the initial solutions proposed by the LLM and hence will not be a suitable indicator of a good configuration/search policy. In order to quantify which setting is the most conducive to algorithm search in our predefined configurations we make note of the following:

Table 4: Performance on sparse real-world whale call datasets.

| Algorithm | Run 1 | Run 2 | Run 3 |
|---|---|---|---|
| E1 (Top 5) | $0.90 \pm 0.01$ | $0.64 \pm 0.02$ | $0.756 \pm 0.009$ |
| E2 (Top 5) | $0.92 \pm 0.11$ | $0.654 \pm 0.036$ | $0.754 \pm 0.018$ |
| E3 (Top 5) | $0.61 \pm 0.093$ | $0.784 \pm 0.135$ | $0.87 \pm 0.034$ |

Table 5: Performance on sparse real-world whale call datasets after Optuna tuning.

| Algorithm | Run 1 | Run 2 | Run 3 |
|---|---|---|---|
| E1 (Top 5) | $0.933 \pm 0.034$ | $0.79 \pm 0.012$ | $0.79 \pm 0.024$ |
| E2 (Top 5) | $0.926 \pm 0.12$ | $0.84 \pm 0.035$ | $0.81 \pm 0.044$ |
| E3 (Top 5) | $0.907 \pm 0.03$ | $0.842 \pm 0.11$ | $0.93 \pm 0.02$ |

Table 6: Performance on dense real-world whale call datasets.

| Algorithm | Run 1 | Run 2 | Run 3 |
|---|---|---|---|
| E1 (Top 5) | $0.81 \pm 0.013$ | $0.56 \pm 0.073$ | $0.60 \pm 0.00$ |
| E2 (Top 5) | $0.76 \pm 0.04$ | $0.69 \pm 0.09$ | $0.50 \pm 0.02$ |
| E3 (Top 5) | $0.72 \pm 0.06$ | $0.70 \pm 0.023$ | $0.78 \pm 0.023$ |

Table 7: Performance on dense real-world whale call datasets after Optuna tuning.

| Algorithm | Run 1 | Run 2 | Run 3 |
|---|---|---|---|
| E1 (Top 5) | $0.806 \pm 0.005$ | $0.544 \pm 0.08$ | $0.654 \pm 0.013$ |
| E2 (Top 5) | $0.77 \pm 0.025$ | $0.79 \pm 0.035$ | $0.6 \pm 0.1$ |
| E3 (Top 5) | $0.84 \pm 0.015$ | $0.7 \pm 0.04$ | $0.78 \pm 0.008$ |

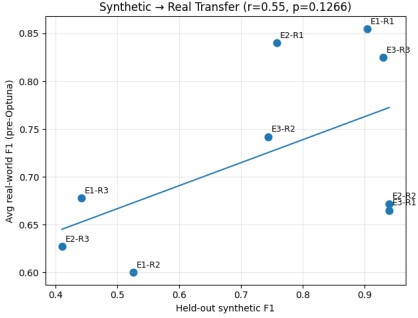 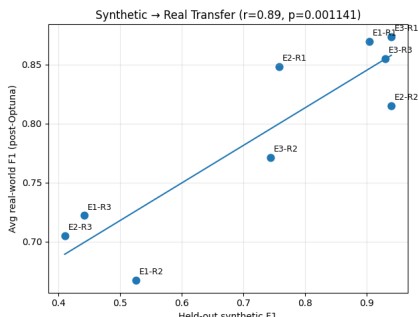

Figure 2: Correlation between held-out synthetic F1 and real-world F1 pre and post Optuna tuning. **Left:** Pre-tuning correlation (r = 0.55, p = 0.13). **Right:** Post-tuning correlation (r = 0.89, p = 0.001).

i) a good configuration can help improve on heuristics/properties of algorithms found in the initial stages; ii) a good search policy will retain the good properties of high performing algorithms from the earlier stages so the performance should not degrade over time; iii) the search process should be more stochastic/chaotic in the early stages and stabilize over time . We quantify these effects using the following metrics:

$$\Delta_{\text{recovery}} (\uparrow) = \max_{t \in \{8,9,10\}} \text{F1}_t - \max_{t \in \{0,1,2\}} \text{F1}_t$$

$$\Delta_{\text{avg}} (\uparrow) = \overline{\text{F1}}_{\text{late}} - \overline{\text{F1}}_{\text{early}}$$
$$= \frac{1}{3} \sum_{t \in \{8,9,10\}} \text{F1}_t - \frac{1}{3} \sum_{t \in \{0,1,2\}} \text{F1}_t$$

$$\Delta_{\text{Var}} (\downarrow) = \text{Var}_{\text{late}} - \text{Var}_{\text{early}}$$
$$= \text{Var}(\{\text{F1}_8, \text{F1}_9, \text{F1}_{10}\}) - \text{Var}(\{\text{F1}_0, \text{F1}_1, \text{F1}_2\})$$

The above quantities for each search policy have been highlighted in 8. Qualitatively, the graphs presented in 3 show that the search policy in E3 configuration with warm starts and scheduled curriculum shows more stable convergence compared to E1 and E2. Similar trends for a smooth convergence were observed for E4 compared to E1, E2, or E5. However, in our analysis with limited runs, we didn't notice any hints of the search benefiting from using a composite objective as opposed to just F1 (E4 vs E3).

**Algorithm Diversity Analysis** We characterize each algorithm by a pipeline tuple consisting of prepossessing method, embedding type, and association logic where each component is assigned to a coarse category based on the implemented operations:

| Metric | E3 | E2 | E1 |
|---|---|---|---|
| $\Delta_{\text{recovery}} (\uparrow)$ | $0.04 \pm 0.053$ | $-0.041 \pm 0.137$ | $0.013 \pm 0.025$ |
| $\Delta_{\text{avg}} (\uparrow)$ | $0.21 \pm 0.09$ | $-0.033 \pm 0.058$ | $0.021 \pm 0.036$ |
| $\Delta_{\text{Var}} (\downarrow)$ | $-0.036 \pm 0.025$ | $-0.0026 \pm 0.011$ | $-0.0003 \pm 0.0005$ |
| Diversity $(\uparrow)$ | $9.67 \pm 0.47$ | $8.67 \pm 1.89$ | $4.33 \pm 0.47$ |

Table 8: Early–late curriculum deltas and diversity metrics. Positive $\Delta_{\text{recovery}}$ and $\Delta_{\text{avg}}$ indicate improved late-stage performance, while negative $\Delta_{\text{Var}}$ indicates increased stability.

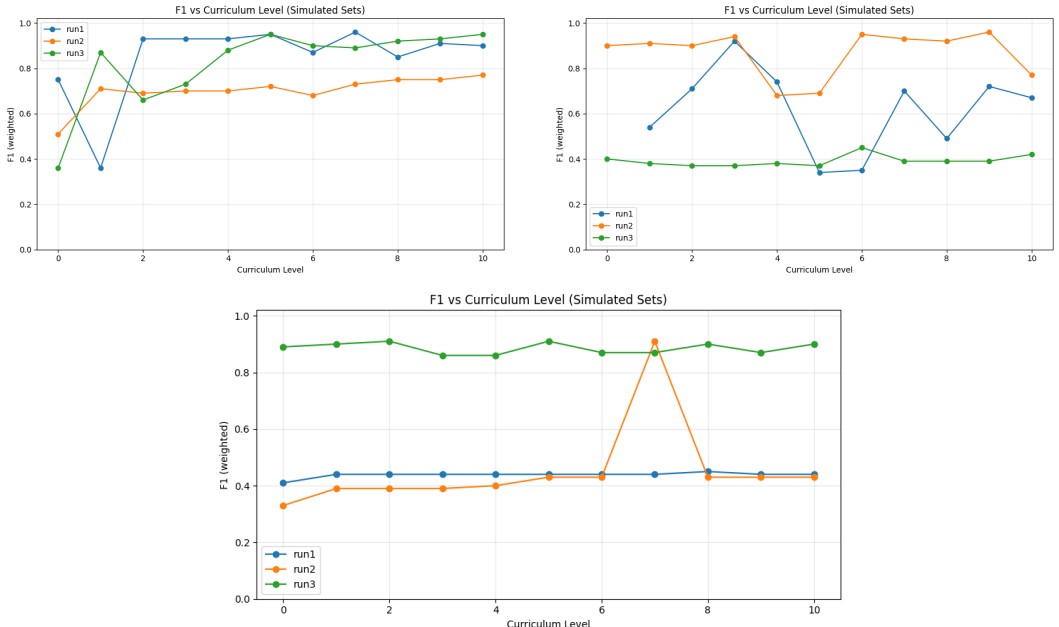

Figure 3: Evolution of F1 on held out synthetic set over iterations under different evolutionary configurations (Refer Table 1). Clockwise from top-left: **Top-Left**: Configuration E3 **Top-Right**: Configuration E2. **Bottom**: Configuration E1. E3 seems to have a more stable convergence overcoming poor, brittle heuristics in the initial stages.

*Preprocessing*: noise reduction (e.g., median / Wiener / Gaussian), normalization (min-max or z-score), log scaling, and resizing method.

*Embedding*: DCT / PCA / CNN (ResNet, ImageNet, etc) / flattened spectrogram / simple statistics.

*Association Logic*: Graph/DSU clustering with TDoA feasibility; thresholding method (percentile, Otsu, or GMM); channel-conflict enforcement.

For evaluating algorithm diversity we look at the number of unique tuples discovered over the sampled algorithms (11 levels/bins) which are reported in Table 8. Configurations E2 and E3 with warm starts show significantly higher diversity in the discovered algorithms compared to E1.

## 6 CONCLUSION

We formulated multi-channel call association as an agentic program synthesis problem, using simulation-based evaluation to discover constraint-aware association pipelines without relying on manually annotated real-world datasets during synthesis. Across held-out synthetic regimes, the discovered algorithms generalize to unseen simulator settings and show meaningful transfer to real whale call datasets, with lightweight Optuna tuning further improving real-world performance under domain shift. These results indicate the feasibility of scalable, automated multi-sensor association via simulation-driven agentic search, and motivate future work on improving simulator fidelity and adaptive regime scheduling.

**Limitations.** Our preliminary evaluations are based on a single LLM backbone and search policy (MCTS). It remains to be seen whether the observed improvements hold under other configurations. In addition, our evaluations rely on a manually specified curriculum of simulation regimes. While this enables scientists to steer the synthesis toward regimes of interest, it also introduces an hyperparameter that may potentially require algorithmic understanding. Future work could reduce this burden and improve robustness by introducing adaptive hyperparameter-selection strategies.

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

## A  SCHEDULED SIMULATION REGIMES

Table 9 summarizes the predefined curriculum used during synthesis. Difficulty increases progressively with call density, array sparsity, and noise levels. Candidate algorithms are evaluated sequentially across levels, and progression to the next level is triggered when performance exceeds a fixed threshold or a maximum iteration budget is reached.

Table 9: Scheduled simulation regimes used during synthesis. Difficulty increases with call density, array sparsity, and noise.

| Level | # Calls | # Receivers | Det. Noise | Spec. Noise |
|-------|---------|-------------|------------|-------------|
| 0 | 60 | 3 | 50000 | 0.00 |
| 1 | 60 | 3 | 45000 | 0.05 |
| 2 | 80 | 4 | 42000 | 0.08 |
| 3 | 80 | 4 | 38000 | 0.10 |
| 4 | 100 | 5 | 35000 | 0.12 |
| 5 | 100 | 5 | 30000 | 0.15 |
| 6 | 140 | 5 | 28000 | 0.15 |
| 7 | 140 | 5 | 24000 | 0.15 |
| 8 | 170 | 7 | 20000 | 0.15 |
| 9 | 170 | 7 | 17000 | 0.15 |
| 10 | 200 | 7 | 15000 | 0.15 |

## B  BASE SEARCH POLICY

We use an MCTS-style tree search to choose which candidate program to modify at each step. The policy first creates 5 initial root drafts. After drafting, with probability 0.4 it selects a debuggable buggy leaf (maximum debug depth 3) and applies a debug action; otherwise it selects a non-buggy node for improvement using UCT:

$$\text{UCT}(i) = \hat{Q}_i + c\sqrt{\frac{\ln N_p}{N_i}}, \quad c = 4.0,$$

where $\hat{Q}_i$ is normalized mean utility of the program at node i, $N_i$ is node visits, and $N_p$ is parent visits.

## C  PROBLEM DESCRIPTION PROVIDED TO THE AGENT

**Whale Call Association Framework**

**Task Overview**

Implement the required entry point for the evaluator:

```
def associate_calls(detections) -> list[str]:
    ...
```

This function should use **arrival times** and **spectrograms of whale calls** to cluster or group detections across multiple receivers that originated from the **same whale at the same time**.

**CRITICAL DESIGN REQUIREMENTS:**

1. **Adaptive Parameters:** Your algorithm must automatically adapt parameters based on the data characteristics. DO NOT use hard-coded thresholds or constants. Instead, compute parameters dynamically from: - Distribution of arrival time differences - Statistics of spectrogram similarities - Receiver geometry constraints (TDoA bounds) - Detection density and temporal clustering patterns

2. **Noise Handling and Preprocessing:** Design appropriate preprocessing to handle noise: - Develop and implement noise reduction/filtering techniques - Consider adaptive preprocessing based on detected noise levels - Ensure preprocessing improves robustness without losing signal information - Preprocessing functions should be implemented as helper functions in `associate_core.py` - Handle spectrograms of varying sizes robustly so concatenation/stacking works without shape errors

3. **Spectrogram Embeddings:** Extract meaningful feature representations from spectrograms for similarity computation: - Load spectrograms from '.npy' files - Apply noise preprocessing/denoising before feature extraction - Choose any embedding strategy that generalizes well; tune

adaptively based on observed performance rather than fixed recipes - Ensure embeddings can be computed on consistently shaped inputs (after your chosen resizing/padding scheme) - Compare embeddings with cosine/Euclidean distance or another suitable similarity metric

**Input Format**

`detections` — a `List[dict]`, one dictionary per detection, with the following keys:

- **Channel** (int): Receiver index (1-indexed) - **ArrivalTime** (float): Arrival time (seconds) - **SxxPath** (str): Path to the '.npy' spectrogram file

From the inputs, only use `ArrivalTime`, `SxxPath`, and `Channel`. Other fields should not be used.

The number of receivers varies across curriculum levels (3–7 receivers). Load receiver positions from `data/receivers.csv`.

**Output Format**

Return a list of group labels (strings), one per detection.

The evaluator compares predicted groupings to ground-truth labels using a **Hungarian-aligned F1 score**.

- Group label values do not need to match ground truth. - Every detection must receive a valid label.

**Constraints**

- At most one detection per channel in each group - Group size can be 1 to $N$ receivers - Favor groups that are: - TDoA-feasible given receiver geometry - Similar in spectrogram embeddings - Prioritize TDoA feasibility first; use spectral cues as secondary evidence

**Design Intent**

Groups must obey **TDoA invariances**:

**Boundedness:**

$$-\frac{\|r_i - r_j\|}{c} \leq \Delta t_{ij} \leq \frac{\|r_i - r_j\|}{c}$$

**Consistency:**

$$\Delta t_{ij} + \Delta t_{jk} + \Delta t_{ki} \approx 0$$

Prefer groups that are mutually consistent and close in spectrogram feature space.

**Receiver Geometry**

Receiver positions are provided in `data/receivers.csv` with columns: - **receiver_id** - **x** - **y**

Load receiver positions as:

```
import pandas as pd
import numpy as np
from pathlib import Path

receivers_path = Path(__file__).parent / "data" / "receivers.csv"
receivers_df = pd.read_csv(receivers_path, index_col="receiver_id")
receivers = receivers_df[["x", "y"]].values
c = 1500.0
```

**Evaluation Metrics**

1. **Weighted F1 Score** (primary metric): Hungarian-aligned F1 score after optimal label matching and composition with secondary metrics. 2. Diagnostics: - Singleton fraction - Mean group size - Duplicate channel violations - TDoA violation rate

Objective: **maximize weighted F1 score**.

**Evaluation Pipeline**

- `associate_calls(detections)` is executed with a 1000-second timeout - Output written to CSV with columns `[ArrivalTime, Group]` - Evaluation performed after Hungarian alignment

**Available Packages**

`numpy`, `pandas`, `torch`, `torchvision`, `scipy`, `sklearn`, and standard library packages. GPU available if present.

**Performance Expectations**

**Primary goal:** maximize F1 score through adaptive, constraint-aware association. **Secondary goal:** computational efficiency.

# D REAL WORLD DATASET DESCRIPTION

This dataset consists of North Atlantic right whale (*Eubalaena glacialis*; henceforth NARW) vocalizations as described in Garcia et al (2025) Garcia et al. (2025). NARW are critically endangered, necessitating effective monitoring to inform conservation efforts. In particular, the "upcall" vocalization is commonly produced by NARW, and thereby frequently used as an indicator of presence Mellinger (2004); Davis et al. (2017). The multichannel audio was recorded by an array of five Marine Autonomous Recording Units (MARUs) positioned within Cape Cod Bay, Massachusetts, a known foraging and resting site for NARW Clark et al. (2010). As upcalls span a range from about 50 to 350 Hz, a sampling rate of 2 kHz was used for recording. We leveraged the analysis pipeline described in Garcia et al. (2025) to construct a realistic, physically-feasible ground-truth dataset for association. Upcalls were first detected by the automated classifier, associated across five channels based on TDoA, and all matches were manually screened to ensure correct association. Next, the associated

sets were used to construct realistic datasets with varying call densities by randomly sampling an emission time while maintaining relative arrival times across channels. In total, the dataset consists of 2025 audio samples representing 405 unique calls across four days (2019-02-20, 2019-03-05, 2019-04-07, 2019-04-25). After random sampling of emission times, variants of the dataset spanned from a duration of 400 seconds (60 calls / minute) to 12800 seconds (about 2 calls/minute). The start times, end times, and magnitude of the spectrograms of all audio samples (computed with a window size of 256 samples, or 0.064 seconds) were provided as input to the agent.

