# OpenReview forum: "Agentic Discovery of Multi-Channel Bioacoustic Association Algorithms"
_ICLR.cc/2026/Workshop/FM4Science — ICLR 2026 Workshop FM4Science Poster_

### Official Review · Reviewer_JQTE · 2026-02-15
**Strong Systems Contribution with Promising Scientific Transfer**

**Rating:** 8
**Confidence:** 3

**Review:**

This paper proposes an agentic program synthesis framework for discovering multi-channel bioacoustic call association algorithms. The system combines LLM-based code generation, Monte Carlo Tree Search (MCTS), physics-based acoustic simulation, and a curriculum-style regime scheduling strategy to progressively increase task difficulty. The goal is to automatically discover algorithms that correctly associate whale calls recorded across multiple hydrophones. The central claim is that combining curriculum-scheduled simulation with agentic code search enables discovery of algorithms that generalize from synthetic acoustic regimes to real-world whale datasets.

---

### Official Review · Reviewer_RKR4 · 2026-02-19

**Rating:** 6
**Confidence:** 3

**Review:**

### Evaluation

**Originality:**
The paper introduces an integration of several ideas: (1) formulating a core bioacoustics problem (multi-channel association) as a program synthesis task for an LLM-based agent, (2) using a curriculum of simulation regimes (scheduled by difficulty) to guide the search, and (3) treating the simulator scheduler as a first-class component to avoid overfitting to a fixed synthetic distribution.

**Quality:**
The experimental design is multifaceted. It compares multiple configurations (E1-E5) to isolate the effects of warm starts, regime scheduling, and objective composition. However, the number of runs (three) is relatively small for drawing strong conclusions about configuration superiority, and some comparisons (e.g., E3 vs. E4 on composite objective) are inconclusive.

**Clarity:**
The paper is generally well-written, with a clear motivation, problem setup, and method description. However, the main text could benefit from a more concise explanation of the search policy (MCTS) and the regime scheduler.

**Significance:**
The paper makes a meaningful contribution to applied bioacoustics and the broader field of AI-driven scientific discovery.

### Pros

1. Casting multi-channel call association as an agentic program synthesis problem is well-suited to the constraints of the domain.
2. Introducing a curriculum of increasing difficulty during search is a way to guide discovery and avoid brittle heuristics.
3. The use of held-out synthetic data for selection sets a standard for empirical work in this emerging area.
4. The appendix includes detailed problem descriptions, simulator parameters, and evaluation details, supporting reproducibility.

### Cons

1. The experiments use a single search policy (MCTS-based) and a single LLM backbone (Gemini 2.5 Pro). It is unclear whether the findings generalize to other search strategies or models.
2. The regime scheduler is manually specified (Table 9). While this allows expert steering, it introduces a hyperparameter that may be difficult to tune in new domains.
3. With only three independent runs per configuration, the reported means and standard deviations are suggestive but not definitive.
4. The real-world validation uses a single species (North Atlantic right whale) and a modest number of calls (405 unique calls). Broader validation across species and recording conditions would strengthen claims of generalizability.
5. The paper does not compare against a state-of-the-art deep learning method trained on the real dataset.

---

### Official Review · Reviewer_kmTc · 2026-02-19
**Promising Application of Agentic Program Synthesis to Bioacoustics**

**Rating:** 6
**Confidence:** 3

**Review:**

### Summary

This paper formulates multi-channel bioacoustic call association — determining which detections across a sparse array of receivers originated from the same source — as an agentic program synthesis problem. The core idea is to use an MCTS-based LLM agent (building on AIDE, Jiang et al. 2025) to iteratively discover association algorithms evaluated on synthetic data from a physics-inspired simulator. The key methodological contribution is a **regime-scheduled simulation orchestrator** that progressively increases difficulty (call density, noise, array geometry) during synthesis, combined with warm-start initialization between regimes. Discovered algorithms are pruned on held-out synthetic data, lightly tuned with Optuna on a small real-world North Atlantic right whale dataset, and evaluated on sparse and dense variants of that dataset.

### Pros

**1. Well-motivated applied problem.** Multi-channel call association is a genuine bottleneck in passive acoustic monitoring, and the paper makes a compelling case that it is under-served by existing ML methods. The problem is real, important for conservation, and the constraint structure (TDoA feasibility + spectral similarity) makes it a natural fit for programmatic search rather than end-to-end learning.

**2. Sensible framing as agentic program synthesis.** The idea that agentic AI can search over the combined space of preprocessing, feature extraction, and constrained grouping is appealing. Unlike standard metric learning, this allows hard physical constraints (TDoA bounds, channel uniqueness) to be incorporated directly in the discovered algorithms. The three-stage pipeline (synthesis → pruning → target tuning) is clean and practical.

**3. Regime scheduling is a reasonable contribution.** Treating simulator difficulty scheduling as a first-class design choice — rather than evaluating on a single fixed regime — is a sensible idea. The analogy to curriculum learning is appropriate, and the ablation evidence (Table 8, Figure 3) showing that E3 (scheduled regimes + warm starts) exhibits better $\Delta_{\mathrm{avg}}$ (+0.21) and lower $\Delta_{\mathrm{Var}}$ (-0.036) compared to E1 and E2 is supportive, if preliminary.

**4. Strong synthetic-to-real correlation post-tuning.** The correlation between held-out synthetic F1 and post-Optuna real-world F1 ($r = 0.89$, $p = 0.001$, Figure 2 right) is a useful practical finding, suggesting that synthetic evaluation can serve as a proxy for real-world algorithm ranking.

**5. Honest about limitations.** The authors acknowledge single-LLM dependency, manual curriculum design, and limited runs. The discussion is candid.

### Cons

**1. Limited methodological novelty.** The paper's primary contribution is regime scheduling with warm starts on top of AIDE's existing MCTS framework. While this is sensible engineering, it is a relatively incremental addition to the agentic program synthesis literature. Curriculum learning for improving search/training is well-established (Bengio et al., 2009), and the scheduling here is entirely manual and static (Table 9). Compared to recent work like AlphaEvolve (Novikov et al., 2025), which introduced population-based evolutionary strategies, multi-objective optimization, and meta-prompt evolution, the technical contribution here is modest. The paper would benefit from a clearer articulation of why this specific combination is non-obvious for the bioacoustics domain.

**2. Very small-scale real-world evaluation.** The real-world dataset consists of only 405 source calls from a single species (NARW) at a single site (Cape Cod Bay), with only 33 used for Optuna tuning and 372 for validation. While the authors rightly note data scarcity is a core challenge, the single-dataset evaluation makes it hard to assess generalizability to other species, array geometries, or acoustic environments. Even within this dataset, the "dense" variant is artificially constructed by compressing timestamps.

**3. High variance across runs undermines conclusions.** The results in Tables 3–7 show substantial run-to-run variation. For example, E1 Run 1 achieves 0.904 on held-out synthetic while Run 3 gets 0.442; E2 ranges from 0.410 to 0.940. With only 3 runs per configuration, it is difficult to make statistically confident comparisons between E1, E2, and E3. The authors acknowledge this ("such performance metrics given a small number of runs are highly dependent on the initial solutions proposed by the LLM") but then still draw conclusions from aggregated metrics. The $\Delta$ metrics in Table 8 are themselves averaged over only 3 runs with high standard deviations relative to effect sizes.

**4. Baselines are weak and incompletely described.** B1 (DL Embedding + Clustering) scores only 0.34 on both datasets; B2 (Greedy TDoA) gets 0.45–0.67. B3 uses "GPT-5.2" with 1-shot extraction — but it is unclear what prompt was used, how much effort went into this baseline, or why a single-shot baseline should be competitive. There is no comparison to the existing bioacoustics association pipelines cited in the related work (e.g., Helble et al. 2015; Lapp et al. 2023), nor to a well-tuned cross-correlation + TDoA pipeline. The gap between B3 and the proposed method could partly reflect the difference in compute budget (hundreds of iterations vs. 1-shot).

**5. Missing details on the simulator and discovered algorithms.** The simulator is described at a high level but key details are missing: What is the spectrogram model? How are propagation losses simulated? What source-level distributions are used? The sim-to-real gap is acknowledged but not characterized. Similarly, the paper does not show or analyze any of the discovered algorithms beyond coarse pipeline-tuple categorization. What do these algorithms actually look like? Do they discover known signal-processing heuristics, or genuinely novel strategies? This is a missed opportunity — the interpretability of discovered programs is one of the main advantages of program synthesis over neural approaches.

**6. Workshop fit is tangential.** FM4Science emphasizes foundation models with scientific priors, domain-specific architectures, uncertainty quantification, and hybrid systems. This paper uses an LLM as a code-generation engine within a search loop — the LLM is a tool, not a foundation model for science. The paper does not engage with the workshop's core themes of physical inductive biases in model architectures, multi-modal scientific foundation models, or evaluation protocols for scientific FM. The contribution is closer to "agentic AI applied to a scientific task" than "foundation models for science."

**7. E4 and E5 are underanalyzed.** Configurations E4 (composite objective) and E5 (combined regime, no warm start) are introduced in Table 1 and Table 3 but receive very little discussion. E5 performs poorly (0.38–0.49 F1), but why? E4 appears comparable to E3 on synthetic data but is never evaluated on the real-world dataset. These omissions weaken the ablation story.

### Minor Issues
- The paper references "GPT-5.2" as a baseline (B3) — this should be verified/clarified given the double-blind setting.
- Table 1 mentions E0 but it's not always consistently included in subsequent tables.
- Some notation inconsistencies: "F1$_t$" indices in the $\Delta$ metrics refer to curriculum levels, not timesteps, which is initially confusing.
- The diversity metric (unique pipeline tuples) is interesting but coarse — two algorithms in the same category could still differ substantially.

### Questions for Authors
1. Can you show one or two representative discovered algorithms in full? What do the best-performing pipelines actually compute?
2. What is the compute cost of the synthesis pipeline (LLM API calls, wall-clock time) compared to manual algorithm development?
3. How sensitive is the regime schedule (Table 9) to its specific parameterization? Was any search over schedules performed?

---

### Meta-Review · Area_Chair_1ZLK · 2026-02-28

**Recommendation:** Accept (Poster)
**Confidence:** 4

**Metareview:**

This paper formulates multi-channel bioacoustic call association as an agentic program synthesis problem, combining LLM-based code generation, MCTS search, and curriculum-scheduled physics simulation. Reviewers appreciate the real-world relevance, thoughtful system design, and the regime-scheduling strategy, which shows promising synthetic-to-real transfer. However, methodological novelty beyond prior agentic frameworks is moderate, empirical variance across runs is high, baselines are relatively weak, and real-world validation is limited to a single species and site. While the work represents a promising applied systems contribution, its connection to foundation models for science is indirect.

---

### Decision · Program_Chairs · 2026-03-03

Accept (Poster)